# Carbon-Energy Impact Analysis of Heavy Residue Gasification Plant Integration into Oil Refinery

Slavomír Podolský, Miroslav Variny * 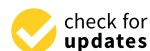 and Tomáš Kurák

Department of Chemical and Biochemical Engineering, Faculty of Chemical and Food Technology,
Slovak University of Technology in Bratislava, Radlinského 9, 812 37 Bratislava, Slovakia;
xpodolsky@stuba.sk (S.P.); tomas.kurak@stuba.sk (T.K.)
* Correspondence: miroslav.variny@stuba.sk

**Abstract:** A gasification plant may partially replace an industrial thermal plant and hydrogen production plant by polygenerating valuable products (hydrogen, power, steam) from low-value materials. Carbon energy analysis is one way of conceptually evaluating such processes. In this paper, the integration of a heavy residue (HR) gasification plant into a mid-size oil refinery (5 million t per year crude processing rate) is conceptually assessed via the comparison of electricity, natural gas and heavy residue consumption, and $CO_2$ emissions. The main purpose of the integration is to reduce the consumption of natural gas currently used for hydrogen production at the expense of increased HR consumption and to achieve a reduction in $CO_2$ emissions. Two case studies with different modes of operation were compared to base case showing that annual reduction of 2280 GWh in natural gas consumption with constant heat and hydrogen production is possible, accompanied with a slight increase in electricity purchase by 28 GWh per year. HR processing in the refinery increases by over 2800 GWh per year. The refinery's $CO_2$ emissions increase by more than 20% (up to 350 kt per year) as a result, while, after incorporating external emissions into the balance, a decrease of more than 460 kt $CO_2$ per year can be achieved. This confirms that the integration of gasification plants within industrial enterprises and clusters has a positive environmental and energy impact and supports the idea of converting low-value material to more valuable products in polygeneration plants. The economics of HR gasifier integration in varying operations under real refinery conditions remain to be explored.

**Keywords:** byproduct upgrade; polygeneration; $CO_2$ emissions; hydrogen; gasification

## 1. Introduction

Modern society is characterized by rapidly increasing crude oil consumption. According to British Petroleum, the consumption of crude oil increased three times (from 30,840 kilo barrels per day to roughly 95,000 kilo barrels per day) from 1965 up to the present day [1]. The processing of such huge amounts of feedstock yields substantial shares of low-valuable by-products. Actual trends in the refining and petrochemical industry are based on searching for suitable solutions to utilize every oil product, especially those with lower value that are often called "bottom of the barrel" [2].

### 1.1. General Principles

Heavy residues (HRs) are by-products of crude oil processing, consisting of large hydrocarbon molecules with heteroatoms embedded in their structure [2]. Due to the stepwise processing of oil, HR also contains a notable volume of heavy metals [3]. Thanks to its composition, HR is a very viscous liquid that is able to flow at temperatures above 150 °C [4]. It is difficult to upgrade it with conventional methods and, thus, it is used as low-price ship fuel or fuel for steam and power plants.

Gasification is one of the promising alternatives for the valorization of low-value by-products to increase their market value [5]. This process can be characterized as the

conversion of liquid and solid by-products and wastes into syngas [6], utilized as a source of hydrogen [7,8]. It is a controlled exothermic reaction of feedstock with suitable gasifying reagents, e.g., air, steam, and oxygen [9,10], consisting of several interconnected steps such as vaporization, pyrolysis, atomization, and chemical fission [11]. Any hydrocarbon-based material can be processed in this way: coal [12,13], tires [14], biomass [15], or plastic materials and wastes [16,17]. The design of a gasification plant depends on processed feedstock, but usually, such a plant consists of a gasifying reactor, syngas-treating section, and an energy-utilizing section [18]. There are many types of gasifying reactors—fluidized-bed, entrained-flow, and fixed-bed reactors—and their use depends on processed feedstock [19]. An entrained-flow reactor is usually used to process HR [20]. This type of reactor can be described as a tubular reactor with a feedstock disperser and gasifying reagents inlet on one side and syngas and unreacted fractions discharge on the other.

Because gasification is an exothermic process, a large amount of technically usable energy is released and can be used for cogeneration purposes [18]. Syngas composition depends on the HR composition and gasifying reagents ratio. In most cases, the largest share is represented by hydrogen and carbon monoxide; other components are carbon dioxide, methane, steam, hydrogen sulfide, and nitrogen [7,21–24], while minor shares of ammonia, gaseous hydrocarbons (ethane, propane, butane), carbonyl-sulfide, and other products of radical reactions can also be observed [3]. The solid phase formed in this process is mainly composed of heavy metals, soot, and tar, which are entrained by high-speed flowing syngas. Therefore, the most important and necessary syngas-treating steps include solid particle separation, cooling, acid gas separation, and the separation of hydrogen [7,25]. Common means of acid gas (mainly hydrogen sulfide) separation include their absorption into amine or hydroxide solution [26,27], while scrubbed sulfur can be recovered using the Claus process [7]. Hydrogen separation requires the inclusion of membrane separation [28], cryogenic separation, or pressure swing adsorption [7,29], which is mostly used thanks to the high purity of the final product and acceptable energy–economic conditions [30].

Standalone gasification plant projects face multiple problems with legislation, social acceptance, and extra capital investment and operation costs to process all by-products and wastes (for example, wastewaters) following environmental standards [31,32]. On the contrary, a gasification plant integrated into an existing enterprise, such as an oil refinery, offers existing infrastructure [18], and can positively affect the operation of the enterprise in terms of energy, economy, and environmental aspects [33] while bringing sustainable solutions to techno-legislative problems [34]. Figure 1 shows the position of a heavy residue gasification plant (HRGP) integrated into the oil refinery.

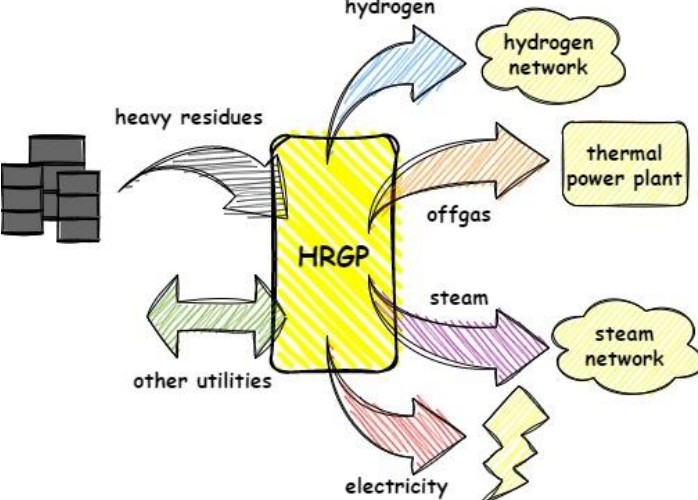

**Figure 1.** Position of heavy residue gasification plant integrated into oil refinery. Source: Own elaboration.

## 1.2. State of the Art

There are multiple studies dealing with the techno-economic or exergy–economic impact of an integrated gasification combined cycle (IGCC) into an existing enterprise, mostly dedicated to biomass, coal, and plastic waste processing.

Ma et al. [35] analyzed the biomass-to-hydrogen process with/without carbon capture and storage (CCS) in terms of energy consumption and efficiency, environmental performance, and techno-economic aspects, but the integration of the process was not analyzed. Al-Rowaili et al. [36] compared methanol production via conventional steam reforming and vacuum residue gasification by the simulation of both processes, stating that gasification is more energy efficient and consumes less energy than steam reforming. IGCC integration into an oil refinery was assumed in this study, but a comparison of the operation before and after the integration was not performed. Sato et al. [37] analyzed the technical and economic aspects of an integrated coke gasifier into a Brazilian oil refinery, but they only evaluated the economic impact. They compared different scenarios of raw material and electricity prices and predicted simple payback periods varying between 7 and 10 years. Berghout et al. [38] analyzed different ways to reduce greenhouse gas emissions in an existing oil refinery (with an already-integrated HR gasifier), as well as in terms of economic benefit. According to the researchers, the most cost-effective pathway is biomass gasification with CCS.

There are several studies that compare and/or analyze IGCC vs. IGCC in combination with another technology to reach higher product qualities. A typical example of such a combination is IGCC coupled with the steam reforming of natural gas, as analyzed by Ahmed et al. [39] and Al-Qadri et al. [40]. Ahmed et al. [39] compared two cases: case 1 represented the integrated entrained-flow gasifier of law rank coal; case 2 was represented by the same type of gasifier coupled with natural gas steam reforming. Al-Qadri et al. [40] simulated and compared the same cases as Ahmed et al. while using waste tires as gasifier feed. In both studies, higher and more efficient hydrogen production was obtained in case 2 according to the techno-economy analysis; the considerable economic benefits of case 2 were also pointed out.

Other studies optimized types and purities of the gasifying agents used in IGCC or IGCC coupled with additional technology to achieve more energy-efficient production. For example, Yeoh and Hui [41] optimized an air separation unit, analyzing and comparing the most favorable oxygen concentration in air. They showed that the gasification of coal is most thermally efficient at an oxygen concentration of 45.5%. Santiago et al. [42] compared the influence of the amount of two different gasifying agents (oxygen and air–steam mixture) on syngas quality, while the syngas produced from oil sludge was combusted in a gas micro-turbine. They showed that oxygen gasification is a more energy-efficient and environmentally friendly syngas and power production method.

The current trend is either to develop new technologies or to find new suitable combinations of known technologies to increase green electricity production and to reduce $CO_2$ emissions. Won et al. [43] identified the optimal configuration of renewable energy sources coupled with biomass gasification. To achieve lower $CO_2$ emissions, different methods of carbon capture and storage have been explored. Khan et al. [44] assessed the operation economics of a biomass-fed fluidized-bed gasifier with an integrated catalytic $CO_2$ adsorber.

As documented by the literature review, many studies analyze and optimize IGCC on its own or find the most suitable conditions for gasification. Other papers deal with biomass or waste gasification. Many studies discuss coupling gasification with other technologies to produce higher-quality products or achieve a more environmentally friendly means of production. On the other hand, how the considered technologies are compared also varies. There are studies analyzing only economic, energetic, or environmental impacts, while others, based on estimation and mathematical modeling, strive to optimize key operation factors to achieve more efficient processes. This highlights the need for a deeper understanding of gasifier integration synergy in a suitable enterprise.

This computational study aims to fill this knowledge gap by assessing the energy and $CO_2$ emissions of the integration of a heavy residue gasification plant (HRGP) into an existing oil refinery. The operation synergies and impact on key refinery production units are estimated and discussed, i.e., an industrial thermal power plant (TPP) and a hydrogen production plant (HPP). The operation features before and after the HRGP integration are compared, focusing on hydrogen and steam production, the consumption of HR and natural gas (NG), HR export outside of the refinery, and electric energy purchase. The amount of $CO_2$ emissions released in both operation states is also evaluated, pointing out the differences in possible approaches. The approach used is generally valid, and it can help decision making in practice and industry management.

## 2. Materials and Methods

### 2.1. Considered System Layout and Feedstock Properties

To design a plant for hydrogen production via heavy residue gasification, studies by Blažek and Rábl [7], Meratizaman et al. [18], Furimsky [20], and Corella and Sanz [25] were considered. Figure 2 shows the HRGP flow chart used in this paper with key streams and equipment designed by our team in a preliminary study [45].

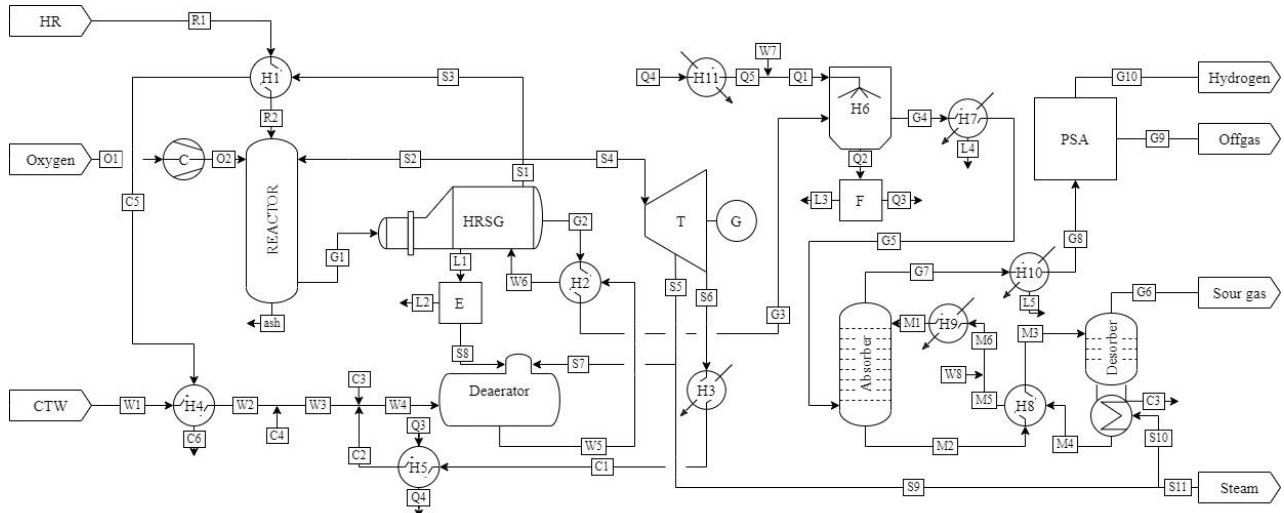

**Figure 2.** Heavy residue gasification plant (HRGP) flow chart: C—oxygen compressor, E—blowdown expander, F—filtration equipment, G—electrical generator, H1–H11—heat exchangers, HRSG—heat recovery steam generator, T—steam turbine; C1–C6—boiler feedwater, G1–G10—syngas, L1–L5—blowdown/wastewater, M1–M6—methyl-diethanolamine (MDEA) solution, O1–O2—oxygen, Q1–Q5—scrubber water, R1–R2—heavy residue, S1–S11—steam, W1–W8—chemically treated water (CTW). Source: Adapted from preliminary study [45].

The properties of HR are closely related to those of crude oil. To model the gasifier, the composition and lower heating value (*LHV*) of HR are required. The typical elementary compositions of HR are presented in Table 1, as reported in the literature.

**Table 1.** HR (heavy residue) composition (mass fractions in %).

| C | H | O | N | S | Ash [1] | Reference |
|---|---|---|---|---|---------|-----------|
| 81.80 | 6.50 | 0.82 | 1.06 | 9.50 | 0.32 | [3] |
| 81.85 | 10.03 | 2.20 | 0.20 | 5.72 | 0.96 | [4] |
| 86.25 | 11.05 | 0 | 0.4 | 2.2 | 0.1 | [21] |
| 85.40 | 11.40 | 0.20 | 0.16 | 2.80 | 0.04 | [46] |
| 84.28 | 10.33 | 0.55 | 0.64 | 3.95 | 0.25 | [47] |

[1] Ash includes heavy metals in HR.

Sulfur is the most variable element, as shown in Table 1. Its content also depends on the crude oil processing method [3,7]. The reaction of sulfur with hydrogen produces hydrogen sulfide, which has acidic properties and could have a corrosive effect on some equipment [48]. Table 2 shows the HR composition and *LHV* applied in gasifier modeling.

**Table 2.** Selected properties of HR, mass fractions in % [21] (*LHV* —lower heating value).

| | | Dry HR | | | | Moisture | LHV [GJ/t] |
|---|---|---|---|---|---|---|---|
| C | H | O | N | S | Ash | | |
| 86.25 | 11.05 | 0 | 0.4 | 2.2 | 0.1 | 0.3 | 40.5 |

### 2.2. Mass and Energy Balance of the Gasifier

According to Furimsky [20], the entrained-flow reactor is most commonly used to process HR by gasification; thus, this type of reactor was chosen for this study. For its mathematical description, a kinetic model was selected [19]. High-pressure superheated steam and compressed oxygen (with 95 vol.% purity) were chosen as suitable gasifying agents. The mass ratio was set to $m_{R2} : m_{O2} : m_{S2} = 1 : 1.1 : 0.35$ according to Blažek and Rábl [7].

The literature review provided several chemical reactions that describe pathways of the complex multistep process of gasification [7,20]. The chemical reactions used in this paper are presented in Equations (1)–(5); their stoichiometry was calculated using HR composition from Table 2.

$$CH_{1.54} + 0.88\,O_2 \rightarrow CO + 0.77\,H_2O \tag{1}$$

$$CH_{1.54} + H_2O \rightarrow CO + 1.77\,H_2 \tag{2}$$

$$CO + H_2O \overset{r_3}{\leftrightarrow} H_2 + CO_2 \tag{3}$$

$$CO + 3H_2 \overset{r_4}{\leftrightarrow} CH_4 + H_2O \tag{4}$$

$$H_2 + S \rightarrow H_2S \tag{5}$$

Equations (1) and (2) represent partial oxidation reactions and Equations (3) and (4) represent equilibrium reactions. "$CH_{1.54}$" in Equations (1) and (2) represents the summary formula based on the elementary composition of HR shown in Table 2 and expresses moles of hydrogen per mole of carbon. The temperature dependence of the reaction rates used in Equations (3) and (5) was calculated via Equations (6) and (7) [24,49], and the temperature dependence of the equilibrium constants was calculated via Equations (8) and (9) [4,49].

$$r_3 = 2700\,exp\left(-\frac{1510}{T}\right)\left[C_{CO}C_{H_2O} - \frac{C_{CO_2}C_{H_2}}{K_{C3}}\right] \tag{6}$$

$$r_4 = 1.585 \cdot 10^7\,exp\left(-\frac{24157}{T}\right)\left[C_{CO}C_{H_2} - \frac{C_{CH_4}C_{H_2O}}{K_{C4}}\right] \tag{7}$$

$$K_{C3} = 0.0265\,exp\left(\frac{3968}{T}\right) \tag{8}$$

$$K_{C4} = T^{-6.567}\,exp\left(\frac{7082.848}{T} + \frac{7.466 \cdot 10^{-3}}{2}T - \frac{2.164 \cdot 10^{-6}}{6}T^2 + \frac{0.701 \cdot 10^5}{2T^2} + 32.541\right) \tag{9}$$

In the case of nitrogen, which is also present in HR, the conversion from atomic to molecular was assumed. The production of nitrogen oxides via high temperature can be neglected [3]. The main syngas components, the concentration of which was calculated, are hydrogen, carbon monoxide, carbon dioxide, steam, nitrogen, sulfur hydroxide, and methane.

For the numerical solution of this reactor model, the following assumptions were implemented:

- In reactions (1) and (2), carbon is converted completely.
- The reactor can be balanced as a CSTR (continuously stirred tank reactor) [50] with the perfect dispersion of inlet HR. The mass balances of each syngas component and the total heat balance of the reactor are presented in Equations (10) and (11).
- Syngas behaves as an ideal gas.

$$\dot{n}_i^{G1} = \dot{n}_i^* + \sum \nu_i r_j V_R \tag{10}$$

where index "*i*" stands for the syngas component and index "*j*" stands for reaction (3) or (4). Symbol "*" stands for the molar flow incurred in reactions (1) and (2).

$$\dot{m}_{R1}\left[c_p^{HR}\left(T_{R2} - T_{ref}\right) + LHV_{HR}\right] + \dot{n}_{O2}h_{O2} + \dot{m}_{S2}\left(\bar{h}_{S2} - 2500\right) = \dot{n}_{G1}(h_{G1} + LHV_{G1}) \tag{11}$$

The solid phase leaving the reactor is neglected in Equation (11) because of its negligible mass flow.

### 2.3. Mass and Energy Balance of other Equipment

In terms of the results from Figure 1, the calculation of every unit of the plant is needed to determine the mass flows and compositions of the final products and the total energy balance of the plant. The mass and enthalpy balance Equations (A1)–(A25) used in HRGP design are presented in Appendix A; the necessary parameters for the design of the whole HRGP are also provided.

### 2.4. Oil Refinery Balance

A typical oil refinery consists of several sub-plants that process different fractions and produce the required products. In this paper, the following oil refinery components were relevant to the conducted analysis: thermal power plant, hydrogen production plant, central steam network, and hydrogen network. Figure 3 shows a simplified refinery layout before and after the integration of the HR gasification plant.

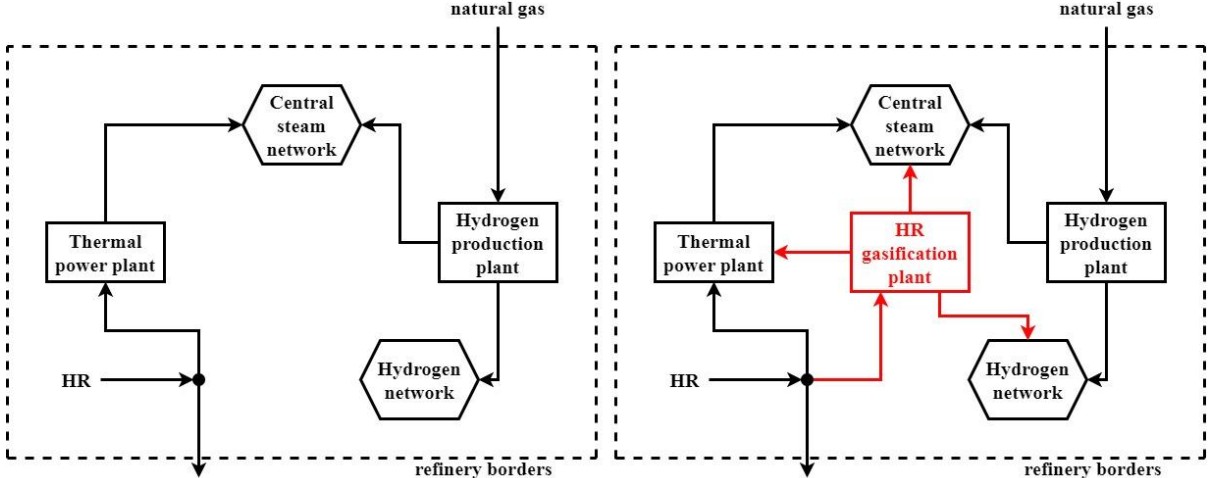

**Figure 3.** Refinery flowchart before and after HRGP integration. Source: Own elaboration.

To analyze the refinery operations before and after HRPG integration, the following factors were compared:

- The consumption of HR in TPP and HRGP;
- The export of HR;
- The consumption of natural gas (NG);
- Steam and hydrogen production;

- The production, consumption, and purchase of electricity;
- $CO_2$ emissions.

The energy assessment is based on the mass and enthalpy balances of each refinery component presented in Figure 2. The calculation premises include an HRGP feed of 60 t/h of HR and estimates of the above factors for a mid-sized oil refinery (Tables 3–5).

**Table 3.** Assumptions applied to refinery balance.

| Parameter | Value | Unit |
|---|---|---|
| Total consumption of electricity | 120 | MW |
| Total amount of HR available | 70 | t/h |
| Summer operation duration | 5000 | h |
| Winter operation duration | 3000 | h |

**Table 4.** Assumptions applied to HPP (hydrogen production plant) balance.

| Parameter | Value | Unit |
|---|---|---|
| Designed hydrogen production | 3.5 | t/h |
| Consumption of natural gas [1] | $4.5\,\dot{m}_{H_2} - 1.75$ | t/h |
| High-pressure steam export [1] | $42 - \left(3.5 - \dot{m}_{H_2}\right)\frac{57}{h_{HP}}$ | t/h |
| Electricity consumption [1] | $0.1\,\dot{m}_{H_2}$ | MW |

[1] In the range of 50 to 100% designed hydrogen production; obtained by non-published data analysis of a real HPP.

**Table 5.** Assumptions applied to TPP; parameters and relations based on unpublished data from a real industrial TPP and study [51].

| Parameter | Value | Unit |
|---|---|---|
| Heat efficiency of steam production | 0.85 | - |
| Specific enthalpy of produced steam | 3.47 | GJ/t |
| Steam export | $0.75\,\dot{m}_{steam}^{produced} - 115$ | t/h |
| Heat in steam production | $2.93\,\dot{m}_{steam}^{export} + 8.95$ | GJ/h |
| Electricity production | $0.23\,\dot{m}_{steam}^{electricity} + 0.08\,\dot{m}_{steam}^{export}$ | MW |

To ensure both reliable and economical hydrogen production, the existence of two smaller HPPs was considered instead of a single larger one. Table 4 shows the assumed design and operation parameters of one such HPP.

Further assumptions need to be introduced to balance the TPP. Figure 4 shows a simplified TPP steam-flow diagram for winter and summer regimes of operation.

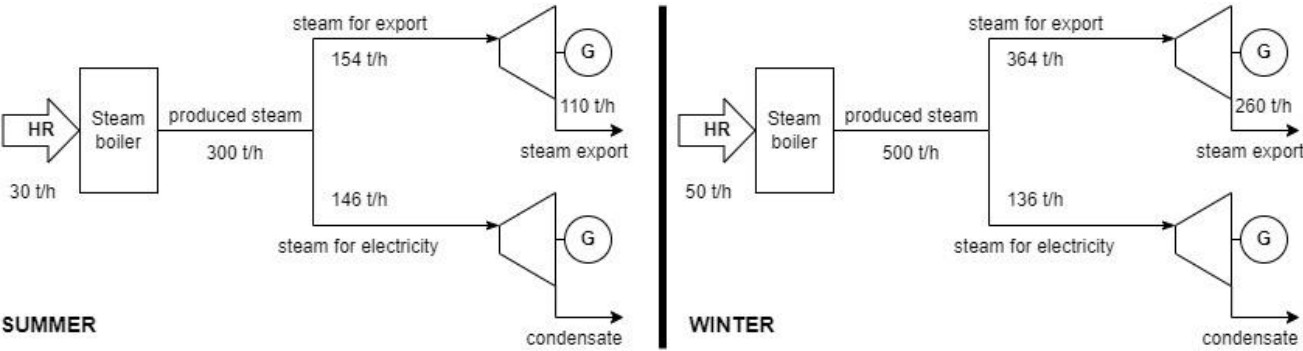

**Figure 4.** TPP (thermal power plant) steam-flow diagram. Source: Own elaboration.

The difference in the steam amount between the "steam for export" and "steam export" streams represents the consumption of steam in the TPP, with the mass flow of condensate considered invariant of season. Mathematical relations describing the TPP operation are shown in Table 5. Offgas obtained by hydrogen separation via PSA is a notable by-product. It is assumed that, due to its sufficient $LHV$, it can be used as additional fuel in the TPP with constant steam production heat efficiency (Figure 3).

HR gasification requires oxygen with high purity (above 95% vol.). Electricity consumption for oxygen separation is calculated as 0.35 $\dot{m}_{O2}$ [MW] [52]. The last of the above-presented factors is the comparison of $CO_2$ emissions; the $CO_2$ emission factors used in the analysis are presented in Table 6.

**Table 6.** $CO_2$ emission factors.

| Source of $CO_2$ | Value | Unit | Note |
| --- | --- | --- | --- |
| HR | 3.16 | t/t | Results from HR elemental composition shown in Table 2 |
| Natural gas | 2.75 | t/t | Considered as pure methane |
| Electricity | 0.102 | t/MWh | Reference [53] |

The HR emission factor is based on its elementary composition shown in Table 2 calculated as tons of $CO_2$ produced by total combustion with pure oxygen per one ton of HR. The NG emission factor is calculated as tons of $CO_2$ produced by the total combustion of pure methane with pure oxygen per one ton of NG.

*2.5. Case Study*

To analyze the impact of integration, the current (base) state has to be defined, and the definition of each state after integration are as follows:

A.    Current state: The total amount of available HR is split between its consumption in the TPP and export outside the refinery. Hydrogen is produced in HPPs, both of which are operated at full load.

B.    New state: A significant amount of HR is processed via gasification; the rest is split between its consumption in the TPP and export outside of the refinery. The total amount of produced hydrogen is the same as in case A; a part of it is obtained via gasification and the rest is produced in one of the HPPs, while the other one is not operated. HRGP exports steam to the refinery during summer and winter. Co-produced offgas is used as additional fuel in the TPP.

C.    New state 2: A significant amount of HR is processed via gasification; the rest is divided between its consumption in the TPP, and export outside the refinery is minimized. The total amount of produced hydrogen is the same as in case A; a part of it is obtained via gasification and the rest is produced in one of the HPPs, while the other one is not operated. HRGP only exports steam to the refinery during winter. Co-produced offgas is used as additional fuel in the TPP.

Case C is a modified version of case B, representing a possible strategic decision of refinery managers aiming at more beneficial, energy-efficient, and environmentally friendly production. During the refinery operation, many non-standard situations may occur (caused by weather changes, changes in feed composition, planned and non-planned shutdowns of plants, and the market demand for different products and their different quality). Switching between the B and C modes of operation can be a favorable solution to make the refinery operation more flexible.

An evaluation of the $CO_2$ amount emitted into air can be carried out via two approaches. The first one is based on the exact calculation of $CO_2$ emitted per year from the refinery via Equation (12) and considers only a part of the $CO_2$ emissions. The other

approach expands the $CO_2$ balance control volume and includes the amount of $CO_2$ emitted per year from HR exported out of the refinery via Equation (13). The first approach yields "refinery emissions", $\dot{m}_{CO_2}^{refinery}$, and the second one provides "total emissions", $\dot{m}_{CO_2}^{total}$.

$$\dot{m}_{CO_2}^{refinery} = \dot{m}_{HR}^{consumed} \cdot e_{HR} + \dot{m}_{NG}^{consumed} \cdot e_{NG} + \dot{m}_{EE}^{purchased} \cdot e_{EE} \tag{12}$$

$$\dot{m}_{CO_2}^{total} = \dot{m}_{HR}^{total} \cdot e_{HR} + \dot{m}_{NG}^{consumed} \cdot e_{NG} + \dot{m}_{EE}^{purchased} \cdot e_{EE} \tag{13}$$

## 3. Results and Discussion

### 3.1. Preliminary Design of HRGP

The plant for heavy residue gasification was designed to process 60 t/h of HR. Its operation is described by Equations (6)–(11) and (A1)–(A25) and by data from Table 2; the results are taken from a bachelor thesis [45]. The main results of this design calculation are presented in Table 7.

**Table 7.** Design calculations of HRGP.

| Medium | Parameter | Value | Unit |
|---|---|---|---|
| HR | Consumption | 60 | t/h |
| | *LHV* | 40.5 | GJ/t |
| Hydrogen | Production | 5.5 | t/h |
| | Purity | 99.9 | vol. % |
| | *LHV* | 141.8 | GJ/t |
| Offgas | Production | 126.4 | t/h |
| | *LHV* | 10.9 | GJ/t |
| Net electricity production | Summer | 18.1 | MW |
| | Winter | 7.7 | MW |
| Low-pressure steam export | Summer | 0 | t/h |
| | Winter | 74.2 | t/h |
| | Enthalpy | 2.87 | GJ/t |
| Oxygen | Consumption | 66 | t/h |

The results for all material flows presented in Figure 2 are shown in Appendix B.

Almost 92 kg of hydrogen per ton of gasified HR is produced, which represents 83% of the hydrogen contained in HR. During gasification, a certain amount of solid phase is produced, which is mainly composed of heavy metals, soot, and tar. This material stream is neglected in the calculations because of its low mass flow.

Thanks to the high temperature of fresh syngas leaving the reactor, 131 t/h of high-quality steam (6 MPa, 460 °C) is produced in the HRSG, 83% of which is used for electricity production, while the rest serves as a gasifying agent in the reactor. More than 75% of the steam inlet into the steam turbine is expanded to low pressure (vacuum) in summer, producing condensing electricity, while this value lowers to 10% in winter when a considerable amount of steam is extracted from the steam turbine and exported out of the HRGP. Besides steam export, steam extraction is also used for desorber heating.

The amount of MDEA solution (flow M1 shown in Appendix B) used for acid gases absorption represents theoretical (minimal) amount for sour gas absorption as the complete saturation of the MDEA solution (full occupation of active centers) is assumed. The produced sour gas (flow G6) can be used for sulfur recovery using the Claus process.

The stepwise pre-heating of boiler feedwater maximizes heat recovery and improves its power production efficiency. Differences between the summer and winter regime operations of the HRGP include a lower cooling water temperature in winter, enabling higher heat power production efficiency, while different electricity production and steam exports are operational differences. As mentioned in Table 7, HRGP serves as power plant during summer and mainly as thermal plant during winter.

### 3.2. Case A

Case A represents reference comparative case. A graphical form of the balances' results from the TPP and HPPs in the refinery is shown in Figure 5. The exact data used in Figure 5 are shown in Appendix C.

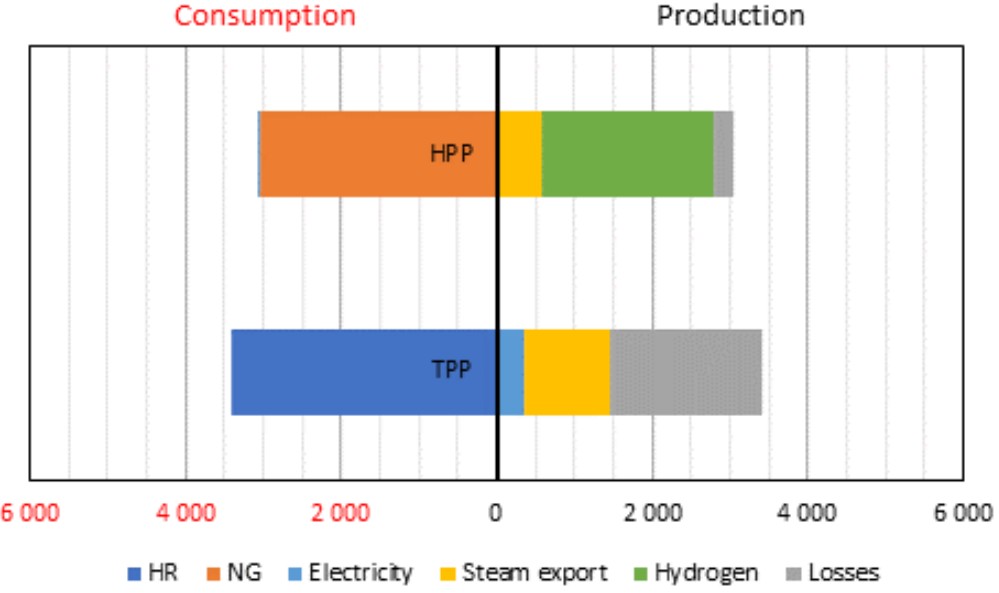

**Figure 5.** Energy diagram of case A (GWh per year). Source: Own elaboration. The term "Consumption" used in Figure 5 (also throughout the analysis) represents energy flows entering the relevant plant and are processed into relevant products leaving each plant, represented by "Production". As presented in Figure 4 (also throughout the analysis), "Losses" represent heat losses coupled to material streams (wastewater, cooling water, etc.) and losses by radiation.

Roughly half of the available HR is currently consumed in the refinery and the rest is sold to external consumers. TPPs can produce only 40% of the electricity consumed in the refinery. The $CO_2$ production analysis shows that the TPP produces 954 kt of $CO_2$ per year and the two HPPs produce 616 kt of $CO_2$ per year. The purchase of electricity contributes to $CO_2$ emissions of 61 kt per year (external emissions).

### 3.3. Case B

HRGP integrated into an oil refinery partially replaces TPPs and HPPs, thereby reducing their throughput. The results of balances after integration are shown in Figure 6. The exact data used to construct Figure 6 are shown in Appendix C. The integration of HRGP causes a reduction in HR consumption in the TPP since the produced HRGP offgas is used as additional fuel for the TPP. After HRGP integration, HR consumption increases to 82% of the available HR. The purchase of electricity increases because of the additional electricity consumption for oxygen separation and because of the lower power production in the TPP. The heat (steam) and hydrogen production rates remain the same. The emissions of $CO_2$ from the TPP decrease to 335 kt $CO_2$ per year. The emissions of $CO_2$ from the HPP decrease to 259 kt $CO_2$ per year due to lower hydrogen production (only one HPP is in operation). The emissions of $CO_2$ from HRGP are 1112 kt $CO_2$ per year as HRGP becomes the main consumer of HR in the refinery. $CO_2$ emissions from the combustion of carbon compounds contained in the HRGP offgas contribute to HRPG emissions (the same applies to case C).

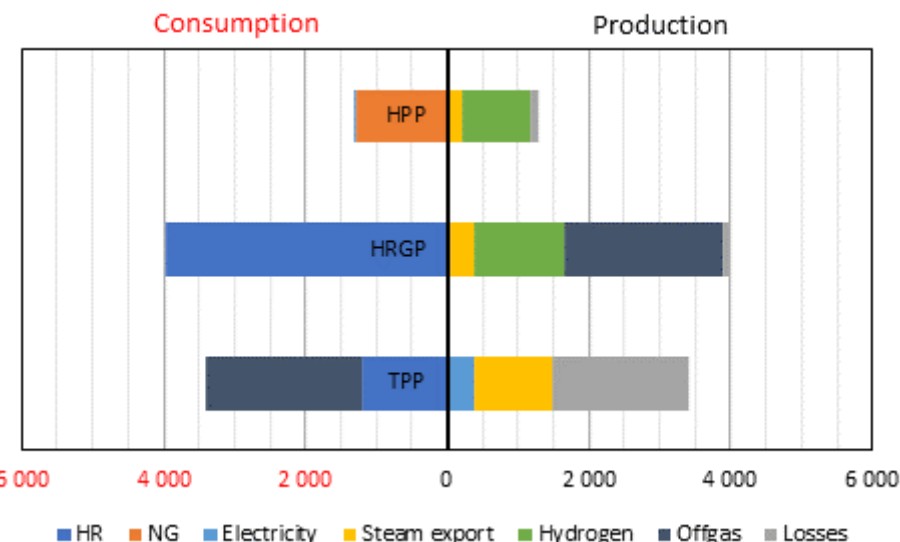

**Figure 6.** Energy diagram of case B (GWh per year). Source: Own elaboration.

*3.4. Case C*

After the modification of the steam export regime from the HRGP to the oil refinery, nearly all HR produced in the refinery is consumed in the HRGP. The results of balances after the integration with this modification are shown in Figure 7 and the source data are shown in Appendix C.

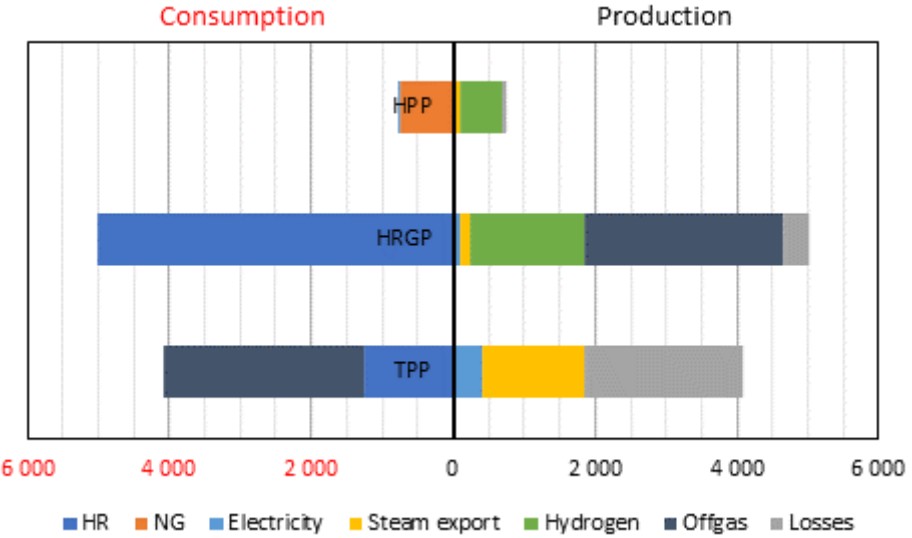

**Figure 7.** Energy diagram of case C (GWh per year). Source: Own elaboration.

As almost all available HR is processed in the HRGP. Further increases in hydrogen production can be observed, leading to a decrease in NG consumption in the HPP of 40% in comparison to case B. The electricity purchase increases by roughly 10% in comparison to case B due to the slightly higher consumption of electricity for oxygen separation. The $CO_2$ emissions from the TPP increase to 350 kt $CO_2$ per year, while those from the HPP decrease to 154 kt $CO_2$ per year and from the HRGP increase to 1408 kt of $CO_2$ per year.

*3.5. Overall Comparison of Cases*

Using the data and dependences presented in Tables 4–7, cases B and C are compared to case A. The results of the mass and enthalpy balance calculations of every component of the oil refinery shown in Figure 3 are presented in Table 8. The stability of the steam and

hydrogen network is ensured by producing the same total amounts of hydrogen and heat (steam) in each analyzed case (56 kt per year of hydrogen and 1682 GWh per year of heat).

**Table 8.** Overall results of state comparison (at constant hydrogen production of 56 kt per year and constant heat production in steam of 1682 GWh per year).

| Category | Factor | Unit (per Year) | A | B | C |
|---|---|---|---|---|---|
| Consumption/Import | HR consumption | kt | 302 | 458 | 556 |
| | Natural gas consumption | kt | 224 | 94 | 56 |
| | Electricity import | GWh | 597 | 683 | 625 |
| Production/Export | Net electricity production [1] | GWh | 363 | 277 | 335 |
| | HR export | kt | 258 | 102 | 4 |
| Emissions | Refinery $CO_2$ emissions | kt | 1633 | 1774 | 1975 |
| | Total $CO_2$ emissions | kt | 2447 | 2098 | 1987 |

[1] Overall "Net electricity production" is calculated as the sum of the differences between electricity production in each considered plant and electricity consumption of each considered plant. Electricity production of HRGP is obtained as the recalculation of designed production (shown in Table 7) to the actual mode of operation (cases B and C); electricity production and consumption of TPP and HPP are presented in Tables 4 and 5. Sum of "Net electricity production" and "Electricity purchase" results in "Total consumption of electricity in the refinery" presented in Table 3.

The comparison of cases A and B shows that the consumption of HR rapidly increased almost 1.5-fold after the HRGP integration. Proportionally to that, both the HR export out of the refinery and the consumption of NG in HPP decreased, as HRGP partially replaces HPP. Due to the lower net electricity production in case B than in case A, the electricity purchase slightly increased.

Regarding $CO_2$ emissions, the higher the consumption of HR, the higher the emissions of $CO_2$ in the refinery. The total $CO_2$ emissions represent the emissions of consumed HR, exported HR, consumed NG, and purchased electricity. Since the sum of the consumed and exported HR is constant and the value of the NG emission factor is higher than for electricity, the reduced consumption of NG leads to a reduction in total $CO_2$ emissions.

The layout in case C was inspired by the HRGP design elaborated in [45]. Adjusting the steam export regime in the HRGP (case C vs. case B) causes almost all of the produced HR to be consumed in the refinery, which leads to a decrease in NG consumption, electricity purchase, and total $CO_2$ emissions.

Due to the TPP fuel switch (HR changed to offgas), additional technical modifications to the steam boiler construction (for example, changes to the burner type) may be needed, leading to additional capital expenses to HRGP commissioning.

The key parameters of the case comparison are the efficiency of hydrogen, heat, and electricity production. Individual efficiencies were obtained by dividing the production of the considered commodity and overall fuel consumption (HR and NG), both expressed in energy units. Table 9 shows the calculated efficiencies of each case.

**Table 9.** Heat efficiencies of commodity production in %.

| Parameter | Case A | Case B | Case C |
|---|---|---|---|
| Hydrogen production | 34.2 | 34.3 | 31.4 |
| Electricity production | 5.6 | 4.3 | 4.8 |
| Heat production | 26.1 | 26.2 | 24.0 |

The reason for heat and hydrogen production efficiencies being either constant or lower is that the heat losses in cases A and B are similar and the heat losses in case C are higher (increased condensing power production). In terms of electricity production efficiency, the electricity consumption necessary for oxygen separation is decisive. To reduce losses, energy optimization is needed, which is a future research topic.

Technically, it is impossible to operate the refinery in only one way due to the many non-standard modes of operation mentioned in Section 2.5. Therefore, it is hard to deter-

mine the exact differences between cases B and C, as each of them has its own benefits in the different analyzed factors.

Besides the changes in the mass and heat balances in individual cases, the amount of $CO_2$ produced in oil refinery depends on the emission factor for electricity, which is different for every country and region. Table 10 shows the corresponding results of refinery emissions analysis before (case A) and after (cases B and C) HRGP integration. The emission factors of several European countries with oil refineries and all continents for the 2021 energy mix were compared in [53].

**Table 10.** Analysis of refinery emissions.

| Country, Region | Electricity Emission Factor [53] (t/MWh) | Before (A) (kt per Year) | After (B) (kt per Year) | Delta (B–A) (kt per Year) | After (C) (kt per Year) | Delta (C–A) (kt per Year) |
|---|---|---|---|---|---|---|
| Slovakia | 0.102 | 1633 | 1774 | 142 | 1975 | 342 |
| Austria | 0.091 | 1626 | 1767 | 141 | 1968 | 342 |
| Czechia | 0.415 | 1820 | 1988 | 169 | 2170 | 351 |
| France | 0.058 | 1606 | 1744 | 138 | 1947 | 341 |
| Germany | 0.354 | 1783 | 1947 | 164 | 2132 | 349 |
| Hungary | 0.201 | 1692 | 1842 | 150 | 2037 | 345 |
| Italy | 0.226 | 1707 | 1859 | 153 | 2052 | 346 |
| Netherlands | 0.331 | 1769 | 1931 | 162 | 2118 | 349 |
| Norway | 0.026 | 1587 | 1723 | 135 | 1928 | 340 |
| Poland | 0.739 | 2013 | 2210 | 197 | 2373 | 360 |
| Spain | 0.169 | 1673 | 1820 | 148 | 2017 | 344 |
| Sweden | 0.012 | 1579 | 1713 | 134 | 1919 | 340 |
| United Kingdom | 0.270 | 1733 | 1889 | 156 | 2080 | 347 |
| Africa | 0.484 | 1861 | 1896 | 157 | 2214 | 353 |
| Asia | 0.539 | 1894 | 2073 | 179 | 2248 | 354 |
| Australia | 0.531 | 1889 | 1945 | 163 | 2243 | 354 |
| Europe | 0.280 | 1739 | 1844 | 151 | 2086 | 347 |
| North America | 0.352 | 1782 | 2036 | 175 | 2131 | 349 |
| South America | 0.204 | 1693 | 2068 | 179 | 2039 | 345 |

The delta between cases C and A fluctuates between 134 and 197 and that between cases C and A between 340 and 360 kt of $CO_2$ per year; thus, it can be concluded that the impact of electricity emission factor on emission analysis is negligible. Marginal emission factor analysis is, thus, not necessary.

The emission factors of Poland and Sweden were applied in the total $CO_2$ emissions analysis due to a large difference in their values; the results are presented in Table 11.

**Table 11.** Comparison of total $CO_2$ emissions.

| Country | Electricity Emission Factor [53] (t/MWh) | Before (A) (kt per Year) | After (B) (kt per Year) | Delta (B–A) (kt per year) | After (C) (kt per Year) | Delta (C–A) (kt per Year) |
|---|---|---|---|---|---|---|
| Slovakia | 0.102 | 2447 | 2098 | −349 | 1987 | −460 |
| Poland | 0.739 | 2827 | 2533 | −294 | 2385 | −442 |
| Sweden | 0.012 | 2393 | 2036 | −357 | 1930 | −463 |

The "Delta" columns in Table 11 confirm that the incorporation of HRGP significantly reduces $CO_2$ emissions, thus contributing to industry decarbonization.

In terms of the Paris Agreement [54], Green Deal [55], and Fit for 55 plan [56], the long-term intention is to produce electricity in a more environmentally friendly way, which results in lowering electricity emission factors to near zero. The fulfilment of this goal is influenced by the actual political situation in each region, which affects the preferred means of electricity production. The results in Tables 10 and 11 can also be interpreted as the trend

of $CO_2$ emission reduction from the actual values to the future "goal" values (with Sweden representing the future goal with emission factors as low as 0.012 t/MWh).

## 4. Analysis Validation and Discussion

Several papers dealing with the technical or techno-economic evaluation of IGCCs were presented in the Introduction. Every study, including this one, analyzes the same factors using different methods; nevertheless, some similarities can be found. Three parameters applied in technical evaluation in this study and the papers mentioned are compared in Table 12.

**Table 12.** Parameters of technical evaluation comparison.

| Parameter | This Paper, Case C | Ma et al. [35] | Al-Rowaili et al. [36] | Sato et al. [37] | Berghout et al. [38] | Al-Qadri et al. [40] |
|---|---|---|---|---|---|---|
| Energy efficiency (%) | 60.2 [1] 35.2 [2] 2.2 [3] 31.8 [4] | 37.88 [2] | 45.7 [1] | 41.2 [3] | 24 [1] | 30.35 [4] |
| Specific electricity consumption (GJ per t $H_2$) | 61.7 | 117.7 | 24.2 [5] | 218.1 [5] | | |
| Specific $CO_2$ emissions (t $CO_2$ per t $H_2$) | 34.8 [7] | 5.4 [5] | 1.25 [5] | | 0.07 [6] | 7.15 [7] |

[1] Overall energy efficiency–energy in form of produced hydrogen, electricity, and heat. [2] Energy efficiency of hydrogen and electricity production of IGCC. [3] Energy efficiency of electricity production of IGCC. [4] Energy efficiency of hydrogen production of IGCC. [5] Specific consumption for IGCC. [6] Due to co-integrated carbon capture and storage plant (CCS). [7] Considered specific emissions of IGCC.

In each study, a different level of refining technology complexity and technologic maturity is analyzed, which is one of the main factors affecting the final values of the compared parameters (for example, co-integrated CCS or steam/electrical engines affect electricity consumption, energy efficiency, and amount of $CO_2$ emitted). In terms of gasification, Ma et al. [35] analyzed the biomass-to-hydrogen process, while Al-Rowaili et al. [36] analyzed vacuum residue gasification coupled with methanol production in comparison to traditional steam methanol production. Sato et al. [37] analyzed an integrated coke gasifier into an oil refinery in a techno-economic way, and Berghout et al. [38] considered a heavy residue gasifier in an environmental way. Finally, the study conducted by Al-Qadri et al. [40] analyzed waste tire gasification coupled with natural gas steam reforming. All of these studies were compared to this paper, analyzing heavy residue gasifier integration into an existing oil refinery using similar energy and environmental evaluation parameters.

Ma et al. [35] compared the efficiency of CCS in a hydrogen production plant via biomass gasification. Similar technology for gasification was also considered in this study and, thanks to that, the estimated energy efficiencies are similar. The main electricity consumer in Ma et al. is CCS, as reflected in the higher specific electricity consumption per ton of produced hydrogen compared to this study.

In the study by Al-Rowaili et al. [36], the same type of gasification reactor and similar syngas outlet temperature and $HR/O_2$ ratio as in this paper were assumed. They compared traditional steam reforming methanol production to methanol production via vacuum residue gasification as an unconventional technique. The constant production of the main product—methanol—was assumed and the accumulation of $CO_2$ emissions was confirmed due to "bottom of the barrel" processing. The subsequent use of captured $CO_2$ in the case of CCS application can lead to a significant reduction in $CO_2$ emissions. This can, under certain circumstances, be a possible solution for emission reduction in case C at the expense of increased electricity purchase, provided the captured $CO_2$ can be used locally.

Sato et al. [37], in their analysis of coke gasifier integration into a Brazilian oil refinery, assumed the same gasification reactor (entrained flow) and the same feed consumption

as in our study. The energy efficiency of the IGCC plant estimated in their study includes the electricity produced in steam and gas turbines (combusting syngas), while this paper only considers steam turbine electricity production, which is probably the reason for the low efficiency of electricity production in the integrated HRGP. They used thermodynamic analysis to prove that IGCC is suitable for electricity production, but an improvement of its integration is needed. According to our study, implementing IGCC has many advantages, for example, the maximization of hydrogen production, the high added value of low-valued by-products, and the operational flexibility of refining processes.

In the analysis by Berghout et al. [38], a large-capacity oil refinery was considered. The IGCC plant was modeled to replace 560 MW of natural gas; an approximately two-fold reduction of NG was observed between cases A and C in our analysis, which is proportionate to the different oil refinery capacities considered in the studies. As highlighted in [38], one of the disadvantages of gasification is excessive water consumption (as a gasifying reagent for steam production). Using only oxygen or air as gasifying reagents partly solves this problem, but it can decrease the $H_2/CO$ ratio in the produced gas and, thus, reduce hydrogen production.

Al-Qadri et al. [40] compared an integrated biomass gasification plant vs. IGCC coupled with natural gas steam reforming. Steam was used as the gasifying agent. Their energy, environmental, and economy analysis results are presented in Table 12 for the IGCC case. The results of their analysis show that IGCC with reforming is more suitable for increasing the $H_2/CO$ ratio in syngas used for methanol production. In comparison to our study, similar hydrogen production efficiencies were observed.

In terms of overall energy efficiency, the presented concept of HRGP integration to yield a polygeneration unit, producing hydrogen, steam, and electricity, is superior to integration layouts focused on producing only one product. Table 12 documents the highest achieved energy efficiency of over 60% in this study, while the reference studies report values of between 30 and 45%. Obviously, incorporating an additional CCS unit would consume a part of the produced energies and would lower the overall energy efficiency, as indicated by that of less than 25% reported by Berghout et al. [38] for a system equipped with CCS.

Regarding all mentioned studies, different coupled technologies were analyzed by different approaches; both considering and omitting CCS are compared. Thanks to CCS, lower $CO_2$ emissions are obtained in several reference studies compared to this study. To reduce $CO_2$ emissions as much as possible, a combination of several technologies (for example, HRGP, CCS, renewable energy sources) can be a suitable solution.

Indirect $CO_2$ emissions (produced in power sources balancing the changed power consumption of the industrial enterprise) have a certain impact on the total $CO_2$ balance, with the impact being dependent on power consumption change and electricity emission factors. In the presented HRGP integration layout, the resulting purchased power change by the refinery (Appendix C—Table A4) does not exceed +20% compared to the base case. As a result, not even the highest power emission factors considered (see Table 11 Poland: 0.739 t/MWh) cause a significant change in $CO_2$ balance which, for all emission factors considered, indicates an increase in emissions within refinery balance boundaries, but a decrease in emissions in total. However, a different outcome could be obtained by the HRGP focusing on power production (IGCC) with absenting CCS where a much higher change in power purchase by the refinery would be expected. Future electricity emission factors are expected to decrease and Sweden, with its current emission factor of 0.012 t/MWh, can serve as an example of how the $CO_2$ balance could look in the future. As indicated in Tables 10 and 11, a smaller electricity emission factor is beneficial from a $CO_2$ balance point of view, leading to a lower increase in $CO_2$ emissions within the refinery after HRGP integration (Table 10), supporting the total $CO_2$ emission decrease (Table 11).

The performed analysis justifies HRGP integration into oil refineries. A deeper carbon energy analysis should include electricity consumption optimization and the minimization of heat losses, which will significantly contribute to the improved heat efficiencies of the

production of three main commodities: heat, electricity, and hydrogen. Future research can also be aimed at:

- Expanding emission analysis to include other greenhouse gases emitted into air ($NO_x$, $SO_x$);
- Economy analysis focused on HRGP integration payback period evaluation and the minimization of HRGP construction costs.

The coupling of electricity consumption optimization and heat loss minimization with the above aspects would result in the multi-level optimization of HRGP integration into oil refineries.

## 5. Conclusions

The analysis of an HR gasification unit integration into an existing mid-size oil refinery (5 million t per year crude processing) presented in this paper is based on energy impact and carbon emissions estimation. As such a gasification unit partially replaces an industrial thermal plant and hydrogen production plant, these two units together with hydrogen and steam network are considered parts of the oil refinery affected and, thus, are part of the analysis. Two modes of operation after the integration (case B and C) are compared to a base case (case A) before integration. The consumption of HR and NG, the purchase of electricity, and $CO_2$ emissions are the main analyzed factors.

The integration of an HRGP results in a polygeneration process converting low-value refinery by-products into valuable ones (such as hydrogen) at heat, hydrogen, and electricity production efficiencies comparable to the current state. This can, in turn, reduce natural gas consumption almost four times (or, in absolute values, from over 3000 to less than 1300 GWh per year (case B) or even to less than 800 GWh (case C) per year) compared to the current state. An overall energy efficiency of over 60% is achieved due to polygeneration, which is significantly higher than the values reported in reference studies focused on one product (electricity or hydrogen) only. Almost all HR produced by the refinery is processed by the HRGP in case C, saturating over 70% of the hydrogen production of dedicated hydrogen production plants in the base case.

Simultaneously, such integration leads to modestly increased electricity purchase by the refinery of up to +20% and to slightly increased refinery $CO_2$ emissions (up to +10%). However, a decrease in external emissions (expanding balance borders) outweighs this trend, and decreases in the overall $CO_2$ emissions of 15 to 20% (up to 350 kt per year in case B or over 460 kt per year in case C) can be achieved in comparison to the current state. The electricity emission factor is found to play a minor role in $CO_2$ balance, with its lower values reducing the refinery emissions increase and contributing to a higher total emissions decrease. As the electricity emission factors are expected to gradually decrease, the presented HRPG integration layout will become more feasible from a $CO_2$ emissions point of view.

Case C represents a more feasible solution considering the evaluated parameters than case B. However, decision making in optimal refinery operations is far more complex and is based on many more factors. Thus, operating the HR gasifier via case B may represent a better option if HR can (temporarily) be sold for a good price, or if there is a major shift in the refinery's steam balance causing the TPP to consume more HR. It is almost certain that a real integrated gasifier operation would vary among many operation states, out of which cases B and C represent only a fraction. However, if the polygeneration design is adopted, as proposed in this study, an additional gasifier operation flexibility is obtained, which would contribute to its operation feasibility in real refinery operation conditions. The real features of a refinery operation should be incorporated to our future studies dedicated to a deeper analysis of HRPG integration benefits.

This conceptual analysis is a good starting point to techno-economy analysis and deeper multi-level optimization with respect to the minimization of the HRGP integration payback period, construction costs, electricity purchase, and heat losses. Future work will be aimed at the mentioned optimization of integrated gasification plant and an appropriate method of carbon capture and storage to enhance the decarbonization impact.

**Author Contributions:** Conceptualization, M.V.; methodology, M.V. and S.P.; software, S.P.; validation, S.P. and T.K.; investigation, S.P.; data curation, S.P. and T.K.; writing—original draft preparation, S.P.; writing—review and editing, M.V. and T.K.; visualization, S.P.; supervision, M.V.; funding acquisition, M.V. All authors have read and agreed to the published version of the manuscript.

**Funding:** This work was financially supported by the Slovak Research and Development Agency, Grant No. APVV-18-0134 and APVV-19-0170.

**Institutional Review Board Statement:** Not applicable.

**Informed Consent Statement:** Not applicable.

**Data Availability Statement:** All obtained data are provided in the article.

**Conflicts of Interest:** The authors declare no conflict of interest. The sponsors had no role in the design, execution, interpretation, or writing of the study.

## Abbreviations

| | |
|---|---|
| CCS | carbon capture and storage |
| CSTR | continuously stirred tank reactor |
| G | electrical generator |
| HPP | hydrogen production plant |
| HR | heavy residue |
| HRGP | heavy residue gasification plant |
| HRSG | heat recovery steam generator |
| IGCC | integrated gasification combined cycle |
| MDEA | methyl-diethanolamine |
| mol. | molar |
| NG | natural gas |
| PSA | pressure swing adsorption |
| S | summer |
| TPP | industrial thermal power plant |
| W | winter |

## Quantities

| | |
|---|---|
| $\dot{G}$ | molar flow of inert gas, mol/s |
| $N_P$ | electric input power, W |
| $N_V$ | electric output power, W |
| $V_R$ | volume of gasifier, $m^3$ |
| $\dot{n}$ | molar flow, mol/s |
| $c_P$ | specific heat capacity in isobaric conditions, J/kg/K |
| $\dot{m}$ | mass flow, kg/s |
| $h$ | specific enthalpy, GJ/t |
| $C$ | molar concentration, $mol/m^3$ |
| $K$ | constant of equilibria |
| $LHV$ | specific lower heating value, GJ/t |
| $M$ | molar mass, kg/kmol |
| $P$ | pressure |
| $R$ | universal gas constant |
| $T$ | thermodynamic temperature, K |
| $e$ | emission factor |
| $m$ | mass, kg |
| $q$ | molar heat, J/mol |
| $r$ | reaction rate, mol/s |
| $t$ | temperature, °C |
| $w$ | mass fraction, mass % |
| $y$ | molar fraction, mol. % |
| $\varepsilon$ | efficiency of recovery, % |
| $\eta$ | efficiency, % |

| | |
|---|---|
| $\kappa$ | Poisson's constant |
| $\nu$ | stoichiometric coefficient |

### Indices

| | |
|---|---|
| abs. | absorption |
| CW | cooling water |
| des. | desorption |
| EE | electrical energy |
| HP | high-pressure steam |
| iz | isoentropic |
| m | mechanical |
| med. | medium |

### Appendix A

**Table A1.** Mass and enthalpy balance of HRGP equipment (CW—cooling water, HRGS—heat recovery steam generator, PSA—pressure swing adsorption unit).

| Equipment | Equation | No. |
|---|:---:|---|
| Oxygen compressor | $N_P = \frac{\kappa}{\kappa-1}\dot{n}_{O1}RT_{O1}\left[\left(\frac{P_{O2}}{P_{O1}}\right)^{\frac{\kappa-1}{\kappa}} - 1\right]\frac{1}{\eta_m\eta_{iz}}$ | (A1) |
| HRSG | $\dot{m}_{W6} = \dot{m}_{S1} + \dot{m}_{L1}$ | (A2) |
| | $\dot{n}_{G1}(h_{G1} - h_{G2}) = \dot{m}_{S1}\bar{h}_{S1} + \dot{m}_{L1}\bar{h}_{L1} - \dot{m}_{W6}\bar{h}_{W6}$ | (A3) |
| | $\Delta\dot{n}_{H_2S} = \dot{G}(Y_{G5} - Y_{G7}); \Delta\dot{n}_{CO_2} = 0.2\dot{n}_{CO_2}^{G7}$ | (A4) |
| Absorber | $\Delta\dot{n}_{H_2S} + \Delta\dot{n}_{CO_2} + \dot{n}_{M1}(Y_{H_2S} + Y_{CO_2}) = w_{MDEA}^{M1}\dot{n}_{M1}$ | (A5) |
| | $\Delta\dot{n}_{H_2S} + \Delta\dot{n}_{CO_2} + \dot{n}_{M1} = \dot{n}_{M2}$ | (A6) |
| | $\dot{n}_{G5}h_{G5} + \dot{m}_{M1}\left[c_p^{M1}\left(t_{M1} - t_{ref}\right) - 2500(1 - w_{MDEA}^{M1})\right] + q_{abs}^{H_2S}\Delta\dot{n}_{H_2S} +$ $q_{abs}^{CO_2}\Delta\dot{n}_{CO_2} \qquad = \dot{n}_{G7}h_{G7} + \dot{m}_{M2}\left[c_p^{M2}\left(t_{M2} - t_{ref}\right) - 2500(1 - w_{MDEA}^{M2})\right]$ | (A7) |
| Desorber | $\dot{m}_{M3} = \dot{m}_{G7} + \dot{m}_{M4}$ | (A8) |
| | $\dot{m}_{M3}c_P^{M3}\left(t_{M3} - t_{ref}\right) + \dot{m}_{S10}\left(\bar{h}_{S10} - \bar{h}_{C3}\right) =$ $\Delta\dot{n}_{H_2S}h_{H_2S}^{G6} + \Delta\dot{n}_{CO_2}h_{CO_2}^{G6} + \Delta\dot{m}_{H_2O}\left[c_p^{H_2O}\left(t_{G6} - t_{ref}\right) + 2500\right] +$ $\dot{m}_{M4}c_P^{M4}\left(t_{M4} - t_{ref}\right) + q_{des}^{H_2S}\Delta\dot{n}_{H_2S} + q_{des}^{CO_2}\Delta\dot{n}_{CO_2}$ | (A9) |
| PSA | $\dot{n}_{G8} = \dot{n}_{G9} + \dot{n}_{G10}$ | (A10) |
| | $\dot{n}_{H_2}^{G10} = \varepsilon_{H_2}\dot{n}_{H_2}^{G8}; y_{H_2}^{G10} = \frac{\dot{n}_{H_2}^{G10}}{\dot{n}_{G10}}$ | (A11) |
| | $\dot{n}_{CO}^{G10} = \dot{n}_{G10}y_{CO}^{G10}$ | (A12) |
| | $\dot{n}_i^{G9} = \dot{n}_i^{G8} - \dot{n}_i^{G10}$ | (A13) |
| Deaerator | $\dot{m}_{W4} + \dot{m}_{S7} + \dot{m}_{S8} = \dot{m}_{W5}$ | (A14) |
| | $\dot{m}_{W4}\bar{h}_{W4} + \dot{m}_{S7}\bar{h}_{S7} + \dot{m}_{S8}\bar{h}_{S8} = \dot{m}_{W5}\bar{h}_{W5}$ | (A15) |
| Blowdown expander | $\dot{m}_{L1} = \dot{m}_{L2} + \dot{m}_{S8}$ | (A16) |
| | $\dot{m}_{L1}\bar{h}_{L1} = \dot{m}_{L2}\bar{h}_{L2} + \dot{m}_{S8}\bar{h}_{S8}$ | (A17) |
| Steam turbine | $\dot{m}_{S4} = \dot{m}_{S5} + \dot{m}_{S6}$ | (A18) |
| | $\dot{m}_{S4}\bar{h}_{S4} = \dot{m}_{S5}\bar{h}_{S5} + \dot{m}_{S6}\bar{h}_{S6} + \frac{N_V}{\eta_m}$ | (A19) |
| Heat exchangers H1, H2, H4, H5, H8 | $\dot{m}_{med.1}\left(\bar{h}_{med.1}^{out} - \bar{h}_{med.1}^{in}\right) = \dot{m}_{med.2}\left(\bar{h}_{med.2}^{out} - \bar{h}_{med.2}^{in}\right)$ | (A20) |
| Heat exchangers H3, H9, H11 | $\dot{m}_{med.}\left(\bar{h}_{med.}^{out} - \bar{h}_{med.}^{in}\right) = \dot{m}_{CW}\Delta\bar{h}_{CW}$ | (A21) |
| Heat exchanger H7, H10 [1] | $\dot{n}_{Gi} = \dot{n}_{Gi+1} + \frac{\dot{m}_{Lj}}{M_{H_2O}}$ | (A22) |
| | $\dot{n}_{Gi}h_{Gi} = \dot{n}_{Gi+1}h_{Gi+1} + \dot{m}_{Lj}\left(\bar{h}_{Lj} - 2500\right)$ | (A23) |
| Heat exchanger H6 | $M_{H_2O}\left(\dot{n}_{G3} - \dot{n}_{G4}\right) = \dot{m}_{Q2} - \dot{m}_{Q1}$ | (A24) |
| | $\dot{n}_{G3}h_{G3} + \dot{m}_{Q1}\bar{h}_{Q1} = \dot{n}_{G4}h_{G4} + \dot{m}_{Q2}\bar{h}_{Q2}$ | (A25) |

[1] Symbols "$i$" and "$i + 1$" in index stand for gas flow specification; thus, in heat exchanger H7, indices "Gi" and "Gi + 1" represent gas flow G7 and G8, and in heat exchanger H10, indices "Gi" and "Gi + 1" represent gas flow G10 and G11.

**Table A2.** HRGP design parameters.

| Parameter | Value | Unit | Ref. |
|:---:|:---:|:---:|:---:|
| $P_{DES}$ | 200 | kPa | [27] |
| $P_{G9}$ | 110 | kPa | [57] |
| $c_p^M$ | 3.939 | kJ/kg/K | [58] |
| $q_{ABS}^{H_2S}$ 1 | −40.7 | kJ/mol | [59] |
| $q_{ABS}^{CO_2}$ 1 | −57.6 | kJ/mol | [59] |
| $t_{G2}$ | 300 | °C | [7] |
| $w_{MDEA}^{M1}$ | 0.1 | - | [27] |
| $y_{H_2S}^{G7}$ | 20 | ppm | [27] |
| $\varepsilon_{H_2}$ | 75 | % | [57] |
| $\eta_{iz}$(compressor) | 80 | % | [60] |
| $\eta_{iz}$ (turbine) | 80 | % | [61] |
| $\eta_m$(compressor) | 95 | % | [60] |
| $\eta_m$(turbine) | 96 | % | [61] |
| $\kappa$ | 1.4 | - | [52] |

1 $q_{DES} = -q_{ABS}$.

## Appendix B

**Table A3.** Results of HRGP balance [45].

| No. | | $\dot{m}$ (kg/s) | t (°C) | P (kPa) | h (kJ/kg) |
|:---:|:---:|:---:|:---:|:---:|:---:|
| C1 | S | 23.24 | 40 | | 167.45 |
| | W | 3.01 | 35 | | 355.92 |
| C2 | S | 23.24 | 71 | | 295.80 |
| | W | 3.01 | 85 | | 604.68 |
| C3 | | 3.46 | 144 | | 604.68 |
| C4 | | 20.62 | 80 | | 334.92 |
| C5 | | 0.61 | 205 | | 877.50 |
| C6 | | 0.61 | 205 | | 209.26 |
| G1 | | 2.44 | 1384 | 3000 | 45656 |
| G2 | | 2.44 | 300 | 2708 | 9037 |
| G3 | | 2.44 | 136 | 2653 | 4099 |
| G4 | | 2.32 | 90 | 2653 | 2643 |
| G5 | S | 2.27 | 40 | 2600 | 1166 |
| | W | 2.27 | 35 | 2600 | 1019 |
| G6 | | 0.05 | 100 | 200 | 3505 |
| G7 | S | 2.24 | 45 | 2548 | 1309 |
| | W | 2.24 | 40 | 2548 | 1163 |
| G8 | S | 2.24 | 40 | 2497 | 1163 |
| | W | 2.24 | 35 | 2497 | 1163 |
| G9 | S | 1.47 | 40 | 110 | 1164 |
| | W | 1.47 | 35 | 110 | 1164 |
| G10 | S | 0.77 | 40 | 2372 | 1161 |
| | W | 0.77 | 35 | 2372 | 1161 |
| L1 | | 0.73 | 276 | | 1213.90 |
| L2 | | 0.51 | 130 | | 546.31 |
| L3 | | 19.12 | 90 | | 376.94 |
| L4 | S | 0.99 | 40 | | 167.45 |
| | W | 1.01 | 35 | | 146.56 |
| L5 | S | 0.03 | 40 | | 164.45 |
| | W | 0.03 | 35 | | 146.56 |
| M1 | S | 29.40 | 45 | | 177.26 |
| | W | 29.40 | 40 | | 157.56 |
| M2 | S | 30.26 | 51 | | 199.83 |
| | W | 30.27 | 46 | | 180.56 |

**Table A3.** *Cont.*

| No. | | $\dot{m}$ (kg/s) | t (°C) | P (kPa) | h (kJ/kg) |
|---|---|---|---|---|---|
| M3 | S | 30.26 | 80 | | 315.12 |
| | W | 30.27 | 80 | | 315.12 |
| M4 | | 28.95 | 120 | | 473.59 |
| M5 | S | 28.95 | 90 | | 353.01 |
| | W | 28.95 | 85 | | 332.92 |
| M6 | S | 29.40 | 89 | | 353.01 |
| | W | 29.40 | 84 | | 330.88 |
| O1 | | 18.58 | 38.3 | 280 | 35 |
| O2 | | 18.58 | 415.6 | 3000 | 380 |
| Q1 | | 188.98 | 80 | | 334.92 |
| Q2 | | 191.18 | 90 | | 376.94 |
| Q3 | | 172.06 | 90 | | 376.94 |
| Q4 | S | 172.06 | 86 | | 359.60 |
| | W | 172.06 | 89 | | 373.38 |
| Q5 | W | 172.06 | 86 | | 359.65 |
| R1 | | 16.67 | 100 | | 87 |
| R2 | | 16.67 | 200 | | 177 |
| S1 | | 36.34 | 460 | 6000 | 3330.0 |
| S2 | | 5.66 | 458 | 3000 | 3330.0 |
| S3 | | 0.61 | 460 | 6000 | 3330.0 |
| S4 | | 30.07 | 460 | 6000 | 3330.0 |
| S5 | S | 6.83 | 205 | 600 | 2870.8 |
| | W | 27.07 | 205 | 600 | 2870.8 |
| S6 | S | 23.25 | 40 | 7.38 | 2330.2 |
| | W | 3.01 | 35 | 5.63 | 2306.2 |
| S7 | S | 3.38 | 205 | 600 | 2870.8 |
| | W | 2.99 | 205 | 600 | 2870.8 |
| S8 | | 0.23 | 130 | 270 | 2720.7 |
| S9 | S | 3.46 | 205 | 600 | 2870.8 |
| | W | 24.08 | 205 | 600 | 2870.8 |
| S10 | | 3.46 | 203 | 400 | 2870.8 |
| S11 | W | 20.62 | 205 | 600 | 2870.8 |
| W1 | | 6.78 | 20 | | 83.86 |
| W2 | | 6.78 | 34 | | 143.78 |
| W3 | W | 27.40 | 69 | | 287.62 |
| W4 | S | 33.47 | 71 | | 296.95 |
| | W | 33.86 | 78 | | 326.10 |
| W5 | | 37.08 | 130 | | 546.31 |
| W6 | | 37.08 | 205 | | 877.50 |
| W7 | | 16.92 | 20 | | 83.86 |
| W8 | S | 0.45 | 20 | | 83.86 |
| | W | 0.44 | 20 | | 83.86 |

## Appendix C

**Table A4.** Source data for case studies.

| Plant | Parameter | Case A (GWh per Year) | Case B (GWh per Year) | Case C (GWh per Year) |
|---|---|---|---|---|
| | HR consumption | 3402 | 1189 | 1247 |
| | Offgas consumption | | 2228 | 2820 |
| TPP | Electricity production | 368 | 369 | 397 |
| | Steam export | 1103 | 1111 | 1461 |
| | Losses | 1931 | 1937 | 2209 |

**Table A4.** *Cont.*

| Plant | Parameter | Case A (GWh per Year) | Case B (GWh per Year) | Case C (GWh per Year) |
|---|---|---|---|---|
| HPP | NG consumption | 3043 | 1277 | 757 |
| | Electricity consumption | 6 | 2 | 2 |
| | Steam export | 579 | 226 | 92 |
| | Hydrogen production | 2206 | 945 | 611 |
| | Losses | 263 | 108 | 57 |
| HRGP | HR consumption | | 3960 | 5012 |
| | Electricity production | | 46 | 112 |
| | Steam export | | 345 | 129 |
| | Hydrogen production | | 1260 | 1595 |
| | Offgas production | | 2228 | 2820 |
| | Losses | | 81 | 355 |
| Refinery | Electricity purchase | 597 | 683 | 625 |
| | HR export | 2898 | 1151 | 41 |

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
