# Peer review of "Carbon-Energy Impact Analysis of Heavy Residue Gasification Plant Integration into Oil Refinery"

_resources, doi:10.3390/resources12060066_

Round 1

Reviewer 1 Report

Integration of a heavy residues gasification plant into a mid-size oil refinery (5 million t per year crude processing rate) is conceptually assessed via comparison of electricity, natural gas and heavy residues consumption and CO2 emissions in this manuscript. It is an interesting work which aims to reduce consumption of natural gas currently used for hydrogen production and to reach CO2 emissions decrease as well. To further improve the manuscript, some suggestions are proposed as follow.

1) Introduction: the significance of this study needs to be stated more specifically. For example, “The above survey highlights the need for deeper understanding of gasifier integration synergy in a suitable enterprise”. What’s the meaning of the phrase “deeper understanding”? by technical perspective or economical perspective? In addition, the methods for energy and CO2 emissions assessment are encouraged to be discussed in the introduction, since it is the fundamental for this study.

2) It is hard to understand the difference and connection between case B and case C. Both of the two cases are new states, why those two cases are designed, and what’s the similarity, difference and significance of those two cases? Which would be the better choice for future development?

3) Please provide citations for the Eq. (12~36), or derivation process if they are originally obtained in your manuscript. You can put it in the Appendix. As well as the CO2 emission factors listed in Table 7.

4) Present CO2 emission factors are considered in the analysis. What the influence of the CO2 emission factors changing due to the varied energy structure?

5) There are many abbreviations in the manuscript. A list of abbreviation is suggested to be added.

Reviewer 2 Report

The paper titled “Carbon-Energy Impact Analysis of Heavy Residues Gasification Plant Integration into Oil Refinery" is reviewed.   Before making a final decision, I have some suggestions to improve the manuscript

·       The sentence in the abstract is not clear and not specific, lines 14-15“Main purpose of the integration is to reduce consumption of natural gas currently used for hydrogen production and to reach CO2 emissions decrease as well”.

·       What do you mean by “to reach CO2 emissions decrease as well”

·       The author should check all abbreviations in the text. The authors might use an abbreviation table at the beginning or end of the text since lots of abbreviations are used in the text.

·       Heavy residue gasification plant (HRGP) flow chart is generated by the authors or used from different studies. This must be clear in the study.

·       There is no proper literature review section in the study. The authors should split the introduction section into two

·       Conclusion section is weak. The authors should add 100 words more to explain the future scope of your research study and its limitations. 

Reviewer 3 Report

Manuscript ID: resources-2312717
Journal: Resources
Title: Carbon-Energy Impact Analysis of Heavy Residues Gasification Plant Integration into Oil Refinery

1. The abstract can be improved by including the existing challenges, motivations and outcomes of the paper.

2. In the introduction section. the objective of the study needs to be clearly stated. Kindly strengthen the significance of the study. It is weak in its present form.

3 Kindly incorporate the research gap in the literature review. Some literature should be considered and referred.

4. The methodology section is well written.

5. The discussion section needs improvement

6. The conclusion section needs serious improvement. Please talk about the future work briefly in the conclusion section.

Good luck!

Reviewer 4 Report

Diction:Major Revision

Summary

I think that this research to evaluate heavy residue gasification plants in the refining processes in terms of their energetic and carbon dioxide emissions evaluation of technological and process the integration of multiple processes are very interesting. And, I think that  this research to organize comparable technology assessments is important in that process of the world going carbon neutral in the future.

However, the following issues need to be corrected or improved upon in this paper.

1. It is not clear from the manuscript whether this study is based on the results of your experiments or the results of your simulations. Please provide clearer explanation of what you have done.

2. There are parts where the explanation of the research methodology is shortage to the presentation of the results. In particular, there are many part of your paper that is not clear explanation for methodology what are you think done because the comparisons presented in the results are not described in the “Methods” part. Please add an explanation more clearly.

3. Many mathematical formulas are used to evaluate the process. Please add some explanations for the words of the formulas.

4. The name of THIS journal is "Resources". Please change the journal name. If there is an error in your submission, please confirm it with the editorial office.

Each Part

Title

The name of the journal in which you submitted is wrong. Please change it to "Resources".

Fig.1

Did this figure show one of a real plant process or not? This figure is more detailed explanation than Fig. 2, which is shows later.

I think that it would be good to show an abstract figure like Fig.2 before detailed figure like Fig.1

Equations

Please clearly write the definition of each word and unit. Then the large number, it is recommended that a these words table be included at the beginning or end of the book.

Equation(1)-(2)

What is mean CH1.54? Please explain more clearly. This paper is read by not only petroleum refiner professional but also other professional researcher for resouces.

Equation(33) etc.

Left side i, right side i+1, is this correct?" If "i" is a syngas component, then the number of i must be in a certain order to be valid.

And, d the range of i, the minimum value of “i” cannot be explained from this formula, so it must be set as the parameter. Please add a explanation of the definition of the parameter setting during the calculation.

2.5 Case study & 3.2-3.4

Because this part is core of the “Result” section, please provide the more detailed explanations. Please provide a table that the reader could easily compare and see the differences. Also, all comparisons made in the "Result" section should be preliminarily presented in 2.5.

Figure.4-6

Please explain the “Net value”. Also, please change the structure of the figure, it is difficult to understand between “Production” and “Consumption”.

Table.11

If plan C (After) is a bad evaluation result, please clarify the explanation since plan C is based on the improved plan in this study, isn’t it?

Also, If you are evaluating CO2 only that is described t-CO2, however you are evaluated GHG that described t-CO2eq?

Conclusion

Please described what the results of this study obstined, based on the methods used and the results obtained. After that, please write a discussion. I think that the thin version does not explain the results of this study easily undestood to the reader.

Round 2

Reviewer 1 Report

Good job, my questions have been addressed.

Author Response

Thank you very much for your favorable opinion on the revised manuscript.

Reviewer 4 Report

Diction:Minor Revision

Summary:

I think that this research to evaluate heavy residue gasification plants in the refining processes in terms of their energetic and carbon dioxide emissions evaluation of technological and process the integration of multiple processes are very interesting.

I think that this revised version has made the significance and content of the study easier for readers to understand.

However, there are some areas need to be modified. Please improving your paper.

Especially, it is difficult to understand what the numbers in Table.10 represent. If Table.10 is properly explained, I think it could be published.

Each Part

Line number

Line numbers are incorrect, although not related to the final publication. Please note it.

P6 equations

The formulas listed in (A1)-(A25) in the Appendix are written in the manuscript. Please remove them as they are not necessary from the explanation in the manuscript.

Figure.5-7

Please increase the resolution of these figures. If you could it, please add “Net production (or consumption)” mark.

Table.8

Please add a table with the major categories of "Consumption," "Production," and "Total”.

Table.10

What are the numbers listed in this table?

It is not clear from the manuscript; I think CO2 emissions are different from Table.11 because the values are different. Please add explanations in the title or in the manuscript so that we could understand the meaning of the values listed in this table.

Table.12

The compared previous studies are different technologies to this study. Please describe the type of technology as well as the “Reference” in the table so that each method could be identified.
